# Evaluation of Structural Viability of Porcine Tracheal Scaffolds after 3 and 6 Months of Storage under Three Different Protocols

**DOI:** 10.3390/bioengineering10050584

**Published:** 2023-05-12

**Authors:** Alberto Bruning Guimaraes, Aristides Tadeu Correia, Ronaldo Soares da Silva, Elizabete Silva dos Santos, Natalia de Souza Xavier Costa, Marisa Dolhnikoff, Marina Maizato, Idagene Aparecida Cestari, Paulo Manuel Pego-Fernandes, Paulo Francisco Guerreiro Cardoso

**Affiliations:** 1Organ and Tissue Laboratory, LIM 61, Division of Thoracic Surgery, Instituto do Coracao do Hospital das Clinicas HCFMUSP, Faculdade de Medicina da Universidade de Sao Paulo, Sao Paulo 05403-904, Brazil; 2Laboratorio de Poluicao Atmosferica Experimental (LIM05), Departamento de Patologia, Faculdade de Medicina FMUSP, Universidade de Sao Paulo, Sao Paulo 01246-000, Brazil; 3Bioengenharia, Instituto do Coração do Hospital das Clinicas HCFMUSP, Faculdade de Medicina da Universidade de Sao Paulo, Sao Paulo 05403-904, Brazil

**Keywords:** transplantation, trachea, bioengineering, tracheal stenosis, regenerative medicine, biomechanical phenomena, cryopreservation

## Abstract

Tracheal replacement with a bioengineered tracheal substitute has been developed for long-segment tracheal diseases. The decellularized tracheal scaffold is an alternative for cell seeding. It is not defined if the storage scaffold produces changes in the scaffold’s biomechanical properties. We tested three protocols for porcine tracheal scaffold preservation immersed in PBS and alcohol 70%, in the fridge and under cryopreservation. Ninety-six porcine tracheas (12 in natura, 84 decellularized) were divided into three groups (PBS, alcohol, and cryopreservation). Twelve tracheas were analyzed after three and six months. The assessment included residual DNA, cytotoxicity, collagen contents, and mechanical properties. Decellularization increased the maximum load and stress in the longitudinal axis and decreased the maximum load in the transverse axis. The decellularization of the porcine trachea produced structurally viable scaffolds, with a preserved collagen matrix suitable for further bioengineering. Despite the cyclic washings, the scaffolds remained cytotoxic. The comparison of the storage protocols (PBS at 4 °C, alcohol at 4 °C, and slow cooling cryopreservation with cryoprotectants) showed no significant differences in the amount of collagen and in the biomechanical properties of the scaffolds. Storage in PBS solution at 4 °C for six months did not change the scaffold mechanics.

## 1. Introduction

Primary tracheal resection with end-to-end anastomosis in not feasible if the length of the resection exceeds 50% of the trachea in adults and 30% in children [1]. Tracheal transplantation using allografts is an alternative for the surgical treatment of long-segment tracheal diseases when the extent of the resection is not compatible with primary airway reconstruction [2]. Tracheal transplantation using bioengineered grafts has been studied and used based on its non-immunogenicity, biocompatibility, and non-carcinogenicity, and its ability to grow epithelium and to promote angiogenesis. Bioengineered tracheal substitutes are based on a decellularized scaffold, reseeded in a bioreactor with the receptor’s own cells [3].

Biobanks are designed to handle biospecimens for future research and clinical studies [4]. The storage of tracheal decellularized scaffolds in a tissue bank can potentially facilitate the translational research and further clinical use of the scaffolds to produce bioengineered tracheas on demand. The expansion of lung transplant programs worldwide has created an opportunity for harvesting airway segments for this purpose. However, both the tracheal decellularization process and the storage protocols require standardization and adaptation to the local resources, regulatory demands, and testing of biomechanical properties and toxicity before their implementation. There have been many preservation modalities proposed for biological material storage in biobanks using different substances, temperatures, vial types, and kits [5,6]. Nevertheless, the storage methods for tracheal scaffolds have been scarcely investigated and can be critical for scaffold viability and mechanical properties during storage.

The present study focused on the development of a decellularization protocol of swine tracheas and tested different preservation methods of the tracheal scaffolds stored up to six months. Testing of the scaffolds included biomechanics, collagen contents, and cytotoxicity.

## 2. Materials and Methods

### 2.1. Tracheal Procurement and Decellularization

The experimental protocol was approved by the Animal Ethics Committee (CEUA-FMUSP, protocol 172/15). The experiments included 96 porcine tracheas obtained from a butchery. Tracheas were harvested immediately after death from 10-month-old Large White pigs, weighing approximately 100 kg. Tracheas were stored on ice slurry and transported to the laboratory within 24 h.

Upon arrival at the laboratory, the tracheal segments were dissected to obtain segments of similar length (59 ± 8.2 mm) and weight (8.69 ± 1.7 g). The segments were then stored in a solution containing 1% penicillin, 1% streptomycin, and 1% amphotericin (Sigma-Aldrich—A5955, St. Louis, MO, USA) for 12 h overnight in the fridge at 4 °C.

Twelve tracheal segments were tested in natura and the remaining eighty-four underwent a ten-cycle decellularization protocol, as previously described [7].

In short, decellularization of the tracheal segments included 10 cycles of agitation for 48 h with 30 mL of a 2% sodium deoxycholate detergent (Sodium deoxycholate—D6750 500G—Sigma-Aldrich, St. Louis, MO, USA) and ethylenediaminetetraacetic acid 0.02% (EDTA; Sigma-Aldrich, St. Louis, MO, USA) in an incubator at 36 °C and 180 rpm, followed by 3 washings with phosphate-buffered solution (PBS) for 10 min. At the end of this process, the scaffolds were exposed to 30 mL of DNAse solution (30 µg/mL of DNAse I (Roche—04536282001, Sigma, Indianapolis, IN, USA), 3 µL of DNAse + 1 mL of 1.3 mM MgSO_4_, and 2 mM CaCl_2_ (Sigma-Aldrich—A5955, Sigma, St. Louis, MO, USA) at 37 °C for 12 h under constant agitation.

Upon completion of the decellularization protocol, 12 scaffolds were assessed immediately, and the remaining 72 were divided into 3 groups (n = 24 for each group) according to the storage protocol.

### 2.2. Storage Protocols

PBS: The scaffolds were immersed in 30 mL of phosphate-buffered-saline (PBS), prepared in the laboratory in separate 50 mL Falcon tubes, and stored in a fridge at 4 °C.

Alcohol: The scaffolds were immersed in 30 mL of 70% alcohol inside a 50 mL Falcon tube and stored in a fridge at 4 °C.

Cryopreservation via slow cooling medium: Scaffolds were immersed in 30 mL of 90% bovine fetal serum (BFS) plus 10% dimethyl sulfoxide (DMSO, Me_2_SO; Sigma, St. Louis, MO, USA) inside a 50 mL Falcon tube. The contents were slowly and progressively cooled at −1 °C/min until reaching −80 °C. The progressive cooling of the scaffolds was achieved by overnight storage in Nalgene containers placed in a −80 °C freezer. The scaffolds were then immersed in liquid nitrogen at −196 °C. The thawing procedure included a period of 45 min at −80 °C, followed by 10 min of heating at 37 °C in a water bath. The remaining DMSO in the scaffolds was washed out with 30 mL of 0.5 M mannitol, followed by 0.25 M mannitol, and, lastly, by PBS. These tracheal segments were then submitted to storage for 3 and 6 months (Figure 1).

### 2.3. Residual DNA Quantification

The efficacy of the decellularization of the scaffolds was assessed by measuring the residual DNA contents. Samples containing 300 mg of lyophilized porcine scaffold were digested at 55 °C overnight with 333 μL proteinase K solution (3 mL buffer solution, 300 µL 10% sodium dodecyl sulfate, 30 µL proteinase K) (Proteinase K—39950-01—LGC Biotecnologia, Cotia, Sao Paulo, Brasil). This solution was divided into six Eppendorf tubes with 555 µL in each. Then, 222 μL of saturated 6 M NaCl and 777 μL chloroform were added, manually homogenized, and incubated in a shaker for 1 h. The samples were centrifuged at 13,000 rpm at 20 °C for 10 min; 600 μL of DNA in the supernatant was transferred to a new tube, and 600 μL of cold 100% ethanol was added. The tubes were mixed and centrifuged at 13,000 rpm at 4 °C for 15 min; the supernatant was discarded, and 1 mL of 70% ethanol was added, followed by centrifugation in the same conditions. The supernatant was discarded, and the tubes were incubated in the open at room temperature for 60 min. The DNA was diluted in 100 μL of milli-q water, and the samples were heated at 55 °C for 5 min, in order to dissolve the DNA pellets. The samples were vortexed, and 2 μL aliquots were quantified by the spectrophotometer (Thermo Fisher Scientific, NanoDrop one, Waltham, MA, USA) [8]. The mass of DNA from the six tubes were summed and divided by 300, and the result was presented in ng/mg of dry tissue.

### 2.4. Toxicity Threshold of the Detergent and Cytotoxicity of the Scaffold

The cytotoxicity of the scaffold and the cytotoxic threshold of the detergent were assessed using the Cell Count Kit-8 (Sigma-Aldrich, St. Louis, MO, USA). This colorimetric molecular test shows mitochondrial activity, and it is quantified by the light absorbance at the 450 nm wavelength in a spectrophotometer (BioTek, model Elx808, Santa Clara, CA, USA). An initial concentration of 5000 BEAS-2B cells (human bronchial epithelial cell line; BEAS-2B (ECACC 951022433) was seeded in a 96-well plate coated with Collagen I (0.06mg/mL PureCol, Advanced BioMatrix, Carlsbad, CA, USA) and bronchial-epithelial growth media (BEGM; Lonza, Basel, Switzerland). The quantification of the cells was obtained using the Neubauer chamber via the dilution of 10 µL of resuspended cells with 10 µL of trypan blue. After 24 h of seeding, the cell adhesion was ascertained visually by inverted microscopy. A standard curve of cell concentration was obtained by plotting five points in a curve corresponding to 5000, 2500, 1250, 625, 313 cells/well. The quality of the reaction was confirmed by wells with blank, positive, and negative controls.

To determine the cytotoxic threshold of the sodium deoxycholate, progressive concentrations of the detergent (0.0625%, 0.125%, 0.25%, 0.5%, 1%, 2%) were prepared, and the challenge was performed by adding 100 µL of the solutions with different concentrations into the wells containing 5000 cells for 8 h. To determine the cytotoxicity of the scaffold, a homogenate was produced. The scaffolds were submitted to washings with 30 mL of 70% ethanol in an incubator overnight, and then washed again 10 times daily with 30 mL of PBS in the incubator. The samples of the scaffold were cut with a round molded knife with an area of 0.33 cm^2^ (corresponding to the bottom area of one well from a 96-well plate). Two pieces, weighing approximately 75 mg each, were separated for each cytotoxicity test. Samples were milled in the homogenizer (Ultra 80, Ultra Stirrer, Sao Paulo, Brasil) with 1 mL PBS (Sigma-Aldrich, St. Louis, MO, USA), and placed in the centrifuge, and the supernatant was aspirated. The challenge was performed by adding 100 µL of the supernatant into the wells with 5000 cells for 8 h.

For both analyses, after 4 h in the incubator, 10 µL of the water-soluble tetrazolium salt (WTS) (Cell Count Kit 8, Sigma-Aldrich, St. Louis, MO, USA) was added to each well. The WTS remained in the outside of the cell membrane, turning the solution into an orange dye in the presence of any contact with the living cell surface. The supernatant was then transferred to another 96-well plate and quantified using a 450 nm spectrophotometer. Each sample was tested in triplicate. The linear regression obtained in the standard curve was used to calculate the number of cells in each well [9]. The cell viability, after the challenge under different detergent concentrations and challenges with the homogenate, were presented in percentages. The cytotoxicity was considered as the difference of 100%, minus the percentage of cellular viability. Both cytotoxic threshold of the SD and cytotoxicity of the scaffold followed the ISO 10993-1:2009 (Standard, I. Biological Evaluation of Medical Devices), and used 70% viability as the cytotoxicity threshold [10].

### 2.5. Mechanical Properties of the Trachea and Scaffold

The tensile tests on fresh pig tracheas and scaffolds were performed using a standard test equipment (Instron 3365, Norwood, MA, USA) and a data analysis software (Bluehill 3 software). Specimens were cut in the midline of the membranous tracheal wall and flattened. The stress–strain test was performed according to ASTM D638-14 (standard test method for tensile properties of plastics) and were evaluated in both axes of the trachea (transverse and longitudinal) [11]. The samples were obtained from the anterior wall of the trachea and cut in longitudinal and transverse axes with a dog-bone-shaped molded knife size V, described in the norm (Figure 2).

Due to the irregularity of the samples, where the strips of cartilage were often thicker than the connective tissue, the calculation of the cross-sectional area of the specimen included measurements of its weight and density to find the volume (density = weight/volume), thus enabling the determination of the cross-sectional area (volume = area × height). The transverse samples were composed mostly of cartilage, whereas the longitudinal samples were composed of cartilage and dense connective tissue.

The uniaxial stress–strain test was performed at a stretch rate of 10 mm/min in a physiological condition, with the sample immersed in a saline bath at 36 °C to avoid dehydration and enable temperature control.

The software recorded the stress–strain curves and generated a table with the raw data, Young modulus (MPa), maximum load (N), and stress at maximum load (MPa).

### 2.6. Collagen Quantification

A round transverse slice that encompassed one tracheal ring was cut off in each sample in the control, decellularized, and storage groups. The slices immersed in 10% formaldehyde were processed to obtain histology slides stained with picrosirius. The slides were then scanned in the histology slide scanner Scanscope System (Aperoio Technologies, Vista, CA, USA), and 10 photographs (400× power magnification) of the submucosa and previous submucosa were obtained in the CaseViewer software (CaseViewer 2.4, 3DHISTECH, Budapest, Hungary) (Figure 2). The photographs were analyzed using the Image Pro Plus software (Image Pro Plus V9.0, Media Cybernetics, Rockville, MD, USA) [12]. The relative content of collagen fibers was evaluated according to the percentage of areas stained with picrosirius, taking the mean value of the ten sections of each sample, and then the mean value of the twelve samples of each group.

### 2.7. Statistical Analysis

Data are expressed as mean and standard deviation (±SD), or median and interquartile range (IQR) (25–75%). The Shapiro–Wilk and Levene tests were used to verify the normality and homogeneity of variance, respectively. One-way analysis of variance (ANOVA), followed by Dunnett tests, was used to analyze the parametric data. Non-parametric data were analyzed using the Kruskal–Wallis test, followed by the Dunn test. All data were analyzed with the Statistic Package for Social Sciences for Windows program (SPSS 21.0 version) and graphs were created with GraphPad Prism 8 (GraphPad Software, San Diego, CA, USA). The level of significance was set at 5%.

## 3. Results

### 3.1. Scaffold Decellularization

The quality of scaffold decellularization was assessed by means of quantifying the residual DNA. The DNA contents fell from 850 ± 123 ng/mg of dry tissue to 20 ± 8 ng/mg of dry tissue, *p* < 0.001. Effective decellularization is considered when there is less than 50 ng/mg of dry tissue [13,14].

### 3.2. Cytotoxicity Threshold of the Detergent and Scaffold

We determined the cytotoxic threshold of the deoxycholate detergent to be a concentration of less than 0.065% for BEAS-2B cells (Figure 3). The cellular viability under the challenge with the homogenate was 42.7% (Figure 4), denoting that the cytotoxicity of the scaffold was 57.3%.

### 3.3. Determination of Tracheal and Scaffold Mechanical Properties

For the porcine trachea in natura, we found that both axes have different mechanical behaviors, and the transverse axis is stiffer than the longitudinal one. The comparison of the Young modulus showed 8.44 ± 2.87 MPa × 0.94 ± 0.61 Mpa, *p* < 0.001, for the transverse and longitudinal axes, respectively. After the decellularization, we observed a different behavior between the two axes, as compared to the in natura samples. In the longitudinal axis, there was an increase in the maximum load and stress at maximum load. Nevertheless, in the transverse axis, there was a decrease in the maximum load (Table 1).

### 3.4. The Effect of the Storage Protocols on the Scaffolds’ Mechanics

The scaffolds’ biomechanical properties are depicted in Figure 3. The evaluation of the mechanical properties of the scaffold in PBS solution at 4 °C did not show significant differences in the Young modulus, maximum load, and stress at maximum load after three and six months of storage in both axes. After three months of storage in alcohol, we found a significant increase in the stiffness of the scaffolds, which was shown to be reversed in the analysis after six months of storage. The evaluation of the mechanical properties of the scaffold in cryopreservation showed a significant increase in the Young modulus in the transverse axis in comparison between scaffolds stored for zero and six months (Figure 5).

### 3.5. Quantification of Collagen

There was no significant difference in the percentage of collagen in the trachea before and after decellularization (Figure 6). The percentage of collagen in the scaffolds was not different, regardless of the storage protocol tested at 3 or 6 months (Figure 7).

## 4. Discussion

The decellularization protocol was effective, and the comparison between the three different storage protocols for porcine tracheal scaffolds did not show significant differences in the mechanical properties tested, nor in the quantified percentage of collagen.

The decellularization protocol used promoted an adequate decellularization of the porcine trachea. Decellularization of the porcine tracheas used a physical–chemical–enzymatic treatment based on sodium deoxycholate, DNAse, and a chelating agent (EDTA), modified from a previously published method [7]. We suppressed the first 24 h passage in a −80 °C freezer, because a single freeze and thaw cycle did not have an effect on the tissues’ properties [14,15,16]. Sodium deoxycholate is an ionic detergent that solubilizes cellular and nuclear membranes; EDTA is a chelating agent that binds divalent metallic ions and disrupts cell adhesion in the extracellular matrix (ECM); and, finally, DNAse breaks down DNA fragments [14,17,18].

Our preference for sodium deoxycholate ionic detergent is based on its ability to retain the structural proteins necessary for tissue function and its capacity to enable a similar mechanical function in the in natura tissue. In comparison with other detergents, the decellularization with sodium deoxycholate preserves the amount and alignment of elastin fiber and collagen fiber [14].

In order to produce a biological implantable device, it is mandatory to assess its cytotoxicity. The cytotoxic threshold is defined as 70% of cellular viability, obtained in the cytotoxicity assay [10]. We elected to test the cytotoxic threshold of the deoxycholate detergent in a concentration of less than 0.065% for BEAS-2B cells. A previous study tested the cytotoxicity of the sodium deoxycholate in four different human cell lines. The sodium deoxycholate became cytotoxic in concentrations above 0.016%. There were different cell viability behaviors among the cell lines, transducing different resistances of the cells to the detergent, leading to different thresholds: 0.0078% for the human bronchial epithelial cells (HBE); 0.0078% for human mesenchymal stem cells (hMSC); 0.002% for primary human pulmonary vascular endothelial cell line (CBF); and 0.016% for human lung fibroblasts (HLF) [9].

Previous studies have stated the need to wash the scaffold to reduce its cytotoxicity; otherwise, the residual detergent in the ECM would disturb the attachment and growth of the seeded cells [19]. Cebotari et al. produced a scaffold and tested it up to 10 washes with PBS. They found a progressive improvement in cell viability: 50% of cytotoxicity after the third cycle, and 15 ± 3% after the tenth [20]. In our study, even after the 10 washes with 70% ethanol and PBS, the scaffold did not reach the desired threshold of >70% of cell viability in the cytotoxicity assay. We hypothesized that the cytotoxicity observed on the scaffold may be due to a small quantity of residual detergent inside the dense ECM, because, even at very low concentration, the sodium deoxycholate is active and toxic.

We tested the mechanical properties to find out whether the remaining ECM obtained after the decellularization protocol was stiff enough to prevent the airway from collapsing. We used the uniaxial test under the norm of ASTM International, which is the standard testing method for tensile properties of plastics, based upon its reproducibility in other settings.

The trachea is composed of various soft connective tissues, adventitia membranes, mucosa, submucosa, and trachealis muscle. The “D”-shaped tracheal cartilaginous rings are responsible for maintaining the stability of the structure. Our decision to analyze the cartilaginous wall was based on this assumption. We, therefore, analyzed two axes of the wall, the longitudinal and the transverse axes, because they have different compositions. The dog-bone punch used to cut the tracheas produced a composite sample in the longitudinal axis that contained cartilage and dense connective tissue in between the cartilaginous rings. On the other hand, the transverse axis sample was represented by a single cartilaginous ring.

In the porcine trachea in natura, it was found that both axes have different mechanical behaviors, and the transverse axis is stiffer than the longitudinal axis, in terms of the Young modulus. Similar results were published by Hoffman at al. [15]. This biological behavior of the trachea is desirable during the respiratory cycle, because there is a need for some rigidity in the lateral–lateral axis, and elasticity in the longitudinal axis.

In the comparison between the trachea in natura and the decellularized scaffold, we found no significant differences in the Young modulus of the longitudinal and transverse axes. The Young modulus (elastic modulus) relates to the stiffness of the material. Other authors who also used detergent–enzymatic protocols did not find significant differences in the Young modulus in decellularized human tracheas [21], porcine trachea [22], rabbit esophagus [23], or mice diaphragm [24].

Besides the chemical factors, the migration of stem cells through the tissues is influenced by mechanical factors. Ideally, the new environment would keep similar matrix stiffness to allow for proper stem cell migration and differentiation [25,26]. The cells seeded over substrates with similar surface topologies and adhesion protein densities react and show different morphologies as the stiffness of the substrate increases [27]. In the transverse axis, the stress at maximum load (ultimate tensile stress) showed no significant difference after decellularization. Despite a 30% reduction in the maximum load, the biomechanical properties of the cartilage were unchanged after decellularization. Hong et al. also did not find significant differences in the stress at maximum load in rabbit tracheal scaffolds [28].

In the longitudinal axis, our study found a different behavior, where there was a significant increase in the stress at maximum load (ultimate tensile stress) after decellularization. This is rather unusual, as other authors found no significant change, or only a slight decrease, in the mechanical properties related to the decellularization following a detergent–enzymatic protocol in any axis. Testing the longitudinal axis and considering the stress at maximum load, human tracheas showed no significant difference [21], and porcine tracheas showed a slight decrease after repeated detergent–enzymatic cycles [29]. Decellularization following the detergent–enzymatic protocol in mice kidneys resulted in an increase in maximum load, and in the stress at maximum load [30]. An increase in stiffness was also found in a rabbit carotid artery [31]; structural analysis in this study showed that the major components of the ECM were preserved, the elastic fibers appeared intact, and loosening and uncrimping of collagen fibers occurred. The space between collagen fibrils also increased, which may have been associated with an increase in stiffness [31].

The present study protocol generated scaffolds that were mechanically adequate for recellularization and posterior implantation. The Young modulus was not altered, favoring cellular migration and differentiation. The stiffness was preserved in the transverse axis, which would ensure that the graft lumen remains open during the respiratory cycle. Likewise, in the longitudinal axis, the increase in the maximum load and stress at maximum load were features that favored anastomosis under tension.

Biobanks, or biorepositories, are laboratories designed to handle biospecimens to allow future research or clinical studies [5]. The idea of creating a tissue biobank with tracheal scaffolds would facilitate translational research on and clinical use of bioengineered tracheas. The prompt availability of scaffolds of different sizes would accelerate the tissue engineering process and avoid the eventual shortage of organ donors. Furthermore, in the eventuality of an unsuccessful transplant in the future, scaffolds would be readily available for recellularization and implantation.

Elastin and collagen are the main proteins of the ECM. These proteins dictate the biochemical profile biological tissues and impact their mechanical behavior [32]. It has not yet been determined whether the structure of the scaffold fails over time due to its intrinsic degradation or in vivo remodeling.

Storage in PBS in the fridge at 4 °C has widespread use, especially for short-term periods. Porcine tracheal scaffolds were successfully preserved for two months in PBS solution in the fridge and remained suitable for cellular reseeding [33]. However, the ECM in living tissues undergoes remodeling, due to enzymatic or hydrolytic degradation. Dentinal collagen stored in water for 500 days showed that the collagen mesh was sparser, and the interfibrillar space widened from approximately 20 nm to 80 nm as a result of hydrolysis [34].

To understand how the degradation of the ECM of the scaffold influences its mechanical properties and viability, Baiguera et al. proposed the storage of the scaffold under phosphate-buffered solution (PBS) solution at 4 °C for one year. They stated that PBS could reproduce the tissue environment in vitro. Their experiment showed a significant change in the elongation at break, as they compared freshly decellularized scaffolds with scaffolds stored for one year. They also found that the amount of elastin and collagen fibers did not change during this period [35]. The histological analysis of the lung scaffolds preserved with PBS at 4 °C for 3 and 6 months showed progressive loss of lung architecture, resulting in atelectasis that was reversible at 3 months, but irreversible at 6 months [36].

The tracheal scaffolds submitted to one year of storage would be inadequate for clinical use, and were used to try to explain the collapse observed in the implanted grafts. This raises the question of which method would be adequate for the storage of biological scaffolds for tissue engineering purposes and how long the scaffold can be stored for before degrading. Using the PBS storage protocol for rabbit esophageal scaffolds, Urbani et al. analyzed the Young modulus and stress at maximum load of the scaffolds. Both mechanical parameters showed a decrease after six months of storage in PBS. The authors concluded that there was a disarray of collagen fibers after three months of storage in PBS at 4 °C [23].

In our study, the evaluation of the mechanical properties of the scaffold preserved in PBS solution at 4 °C did not show significant differences in the Young modulus, maximum load, or stress at maximum load after three and six months of storage, in either the longitudinal or transverse axes.

We decided to test the storage in 70% alcohol at 4 °C, which is a simple and inexpensive method of storage available in any laboratory. ENT surgeons use 70% alcohol for the storage of cartilage grafts for septoplasty surgery, but the changes that the alcohol promotes in the cartilage are not well studied. The histological analysis comparing alcohol with other kinds of storage (gentamicin, cefazolin, saline, and chloroform) showed a better result for the alcohol group, in which there was less chondrocyte necrosis [37]. After three months of storage, we found a significant increase in the stiffness (Young modulus) of the tracheal scaffolds, which was then reversed at six months. There are scarce data in the literature on the storage of scaffolds in alcohol solution, and we do not have an explanation for this finding at the moment. Further research will be necessary to explain the reversal of stiffness after a longer storage period. Nevertheless, such findings may not rule out alcohol preservation as an alternative for now.

Cryopreservation is a technique developed to allow the preservation of organs and tissues for longer periods. The aim is to be able to recover cell structures and tissue components after thawing. At temperatures of −196 °C in liquid nitrogen, all chemical reactions, biological processes, and physical intra- and extracellular activities are suspended. In non-decellularized tissues, the freezing process causes a 95% loss of intracellular water, which increases the electrolyte concentration in both intra- and extracellular media, causing ice crystal formation in the intracellular spaces that deform and compress cells and destroy intracellular structures. The slow cooling of the tissues induces cell dehydration, preventing crystallization from happening inside the cells. A diffusible cryoprotectant, such as dimethyl sulfoxide (DMSO), has a molecular weight of less than 400, which allows it to easily cross the cell membrane. DMSO forms hydrogen bonds with water, maintaining the liquid state below freezing temperatures. This solute constitutes a solvent for electrolytes, reducing the effect of intracellular concentration. It also blocks the crystallization process by inactivating the condensation nuclei [38]. The use of cryoprotectants in the cryopreservation of scaffolds may be questionable, because the main effect is on preserving cell viability [39]; however, the effects of ice crystal formation inside the ECM may also disorganize the collagen and elastin fibers. On the other hand, avoiding crystal formation and subsequent cracking of the ECM may help to maintain some desirable properties, such as porosity and stiffness [40].

Cryopreservation using a slow cooling protocol is simple, safe, and reproducible, and can be reliably set up in most laboratories or biobanks using basic cryogenic equipment. The slow cooling protocol has the advantage of being less variable than vitrification. Vitrification is another protocol of cryopreservation that avoids ice crystal formation. However, vitrification could be cytotoxic, with minimal changes in the scaffold thickness, the degree of ECM deposition, as well as the cryo-tolerance of cells, leading to the need for constant changes in the protocols [41].

As we evaluated the mechanical properties of the scaffold under cryopreservation, there was a significant increase in the Young modulus between samples stored for zero and six months. Wang et al. evaluated the stress at maximum load of the porcine scaffolds after cryopreservation for three months, and they also found no significant difference [42]. Again, as compared to the storage of rabbit esophageal scaffolds, the Young modulus presented a different behavior [23]. This was probably due to different influences of cryopreservation on smooth muscle (esophagus) and dense connective tissue (tracheal in longitudinal axis).

There was no significant difference in the percentage of collagen in the trachea before and after decellularization. The amount of collagen in the scaffold was not different across storage protocols tested at 3 and 6 months compared to zero months. Usually, there is no change in the number of collagen fibers after decellularization with the present protocol. We acknowledge that a more accurate quantification would require more specific methods, such as immunohistochemistry or immunofluorescence, which were not used in the present study. Previous studies with porcine trachea [29], mice diaphragm [24], and mice kidney [30] confirmed these findings. Cryopreservation after decellularization seems to cause no significant change in the number of collagen fibers in the porcine tracheal scaffolds [42].

The limitations of our study include the cytotoxicity of the scaffold, which will be assessed in future works, using less aggressive decellularization protocols and improved washes. The collagen contents obtained may not be very accurate with the methodology used. The biomechanical characteristics of the scaffolds preserved in 70% alcohol will require further studies to confirm the present findings and perhaps explain the reversal of the stiffness after 6 months. Lastly, despite their morphological and mechanical similarities, swine trachea might not emulate the results in human tracheas.

The production of bioengineered tracheal substitutes offers a research opportunity for complex clinical diseases affecting the central airway. The storage of the tracheal scaffolds in a biobank would facilitate the implementation and diffusion of these bioengineering techniques in future clinical trials.

## 5. Conclusions

The decellularization of porcine tracheas generates a structurally viable scaffold with a preserved collagen matrix suitable for further bioengineering. However, despite the cyclic washings, the scaffolds remained cytotoxic. The comparison among the three storage protocols (PBS at 4 °C, 70% alcohol at 4 °C, and slow cooling cryopreservation with cryoprotectants) did not show significant differences in the amount of collagen, nor in the biomechanical properties of the scaffolds. The simple storage in PBS solution at 4 °C for six months did not change the scaffold’s mechanics.

## Figures and Tables

**Figure 1 bioengineering-10-00584-f001:**
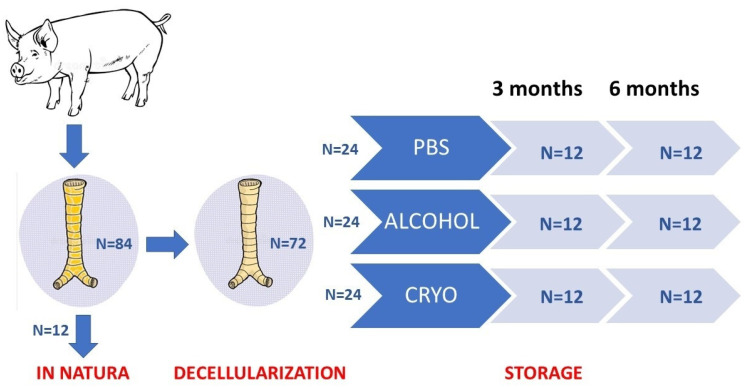
Study design.

**Figure 2 bioengineering-10-00584-f002:**
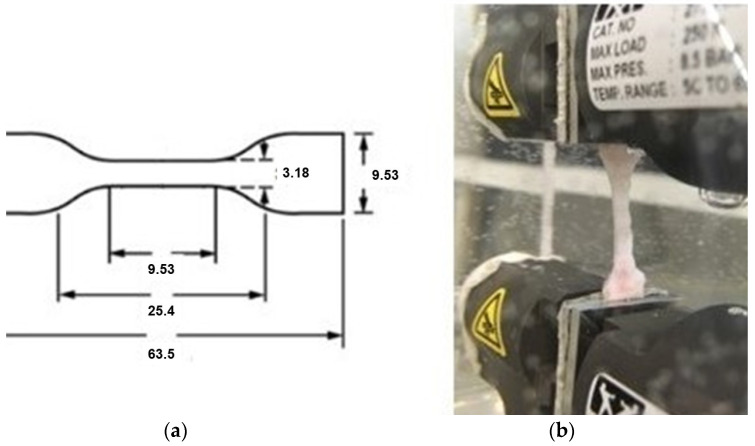
(**a**) Dimensions of the specimen for uniaxial stress-strain test (mm). (**b**) The uniaxial test (longitudinal sample).

**Figure 3 bioengineering-10-00584-f003:**
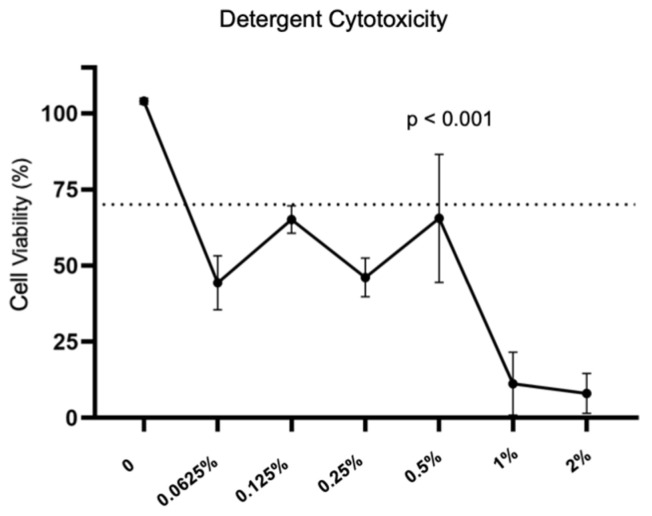
Cytotoxicity threshold of the sodium deoxycholate. The dotted line represents the 70% threshold of cell viability.

**Figure 4 bioengineering-10-00584-f004:**
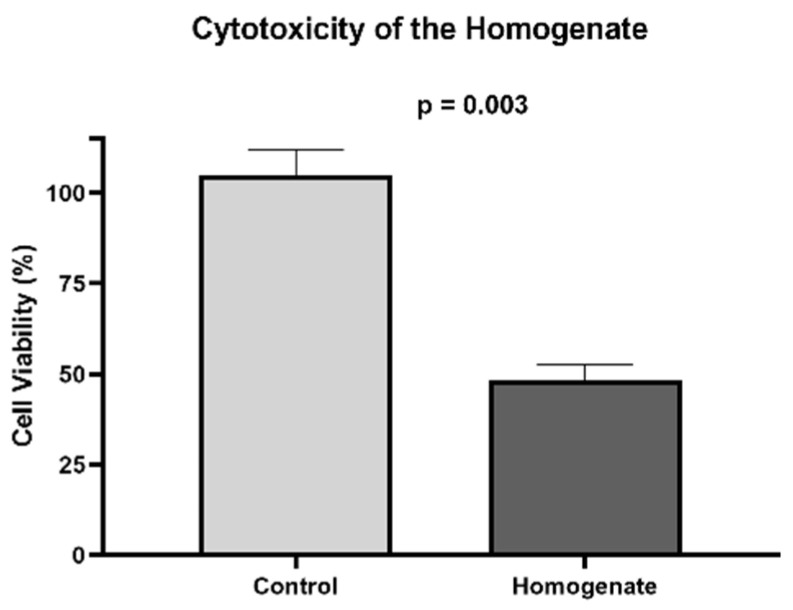
Cytotoxicity of the scaffold.

**Figure 5 bioengineering-10-00584-f005:**
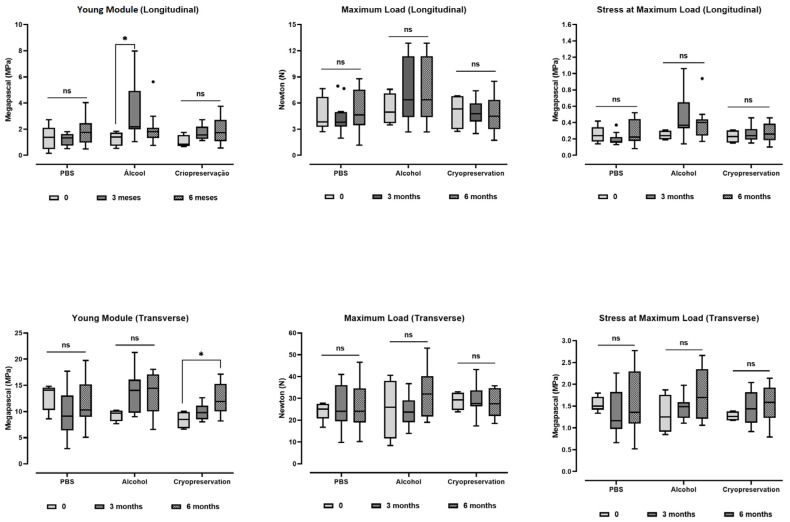
Mechanical properties of the porcine tracheal scaffolds under three different protocols of storage (PBS, alcohol, and cryopreservation) at zero, three, and six months. (*) *p* < 0.05. ns: no significance.

**Figure 6 bioengineering-10-00584-f006:**
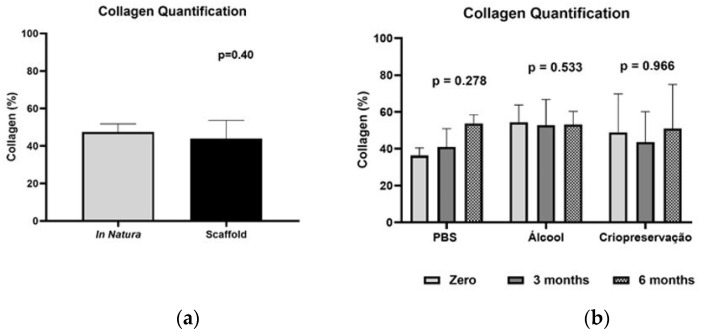
Percentage of collagen fibers (%). (**a**) In natura vs. scaffold; (**b**) Comparison between zero, three, and six months.

**Figure 7 bioengineering-10-00584-f007:**
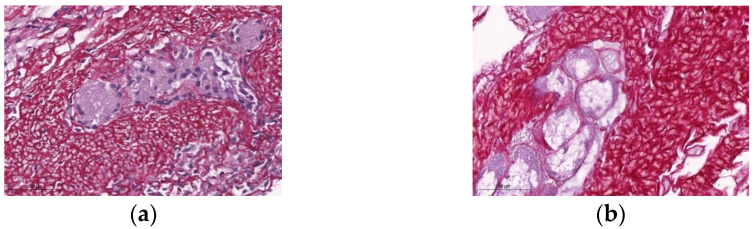
Photomicrographs of histology slides stained with picrosirius for the quantification of collagen fibers obtained from the submucosal layer (400× magnification). (**a**) In natura; (**b**) Decellularized scaffold.

**Table 1 bioengineering-10-00584-t001:** Mechanical properties of the in natura trachea and scaffold.

**LONGITUDINAL**	**In Natura**	**Decellularized**	** *p* **
Young (MPa)	0.94 ± 0.61	1.21 ± 0.67	0.31
Maximum load (N)	3.03 ± 1.47	4.99 ± 1.78	0.009
Stress at maximum load (MPa)	0.14 ± 0.08	0.24 ± 0.07	0.005
**TRANSVERSE**	**In Natura**	**Decellularized**	** *p* **
Young (MPa)	8.18 ± 2.87	10.35 ± 2.68	0.06
Maximum load (N)	36.79 ± 13.39	25.98 ± 7.87	0.02
Stress at maximum load (MPa)	1.65 ± 0.69	1.39 ± 0.28	0.25

## Data Availability

The data presented in this study are available on request from the corresponding author.

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
