# Peer review of "Evaluation of Structural Viability of Porcine Tracheal Scaffolds after 3 and 6 Months of Storage under Three Different Protocols"

_bioengineering, 2023, doi:10.3390/bioengineering10050584_

Round 1

Reviewer 1 Report

In this work the authors have evaluated the effects of storage of decellularized tracheas in a porcine model. They have compared 3 protocols: PBS, alcohol and cryopreservation and have studied the effects at 3 and 6 months. They have not found significant differences in relation to collagen content or biomechanical properties. In my opinion it is a good study necessary because there are very few previous studies. It could be accepted for publication, but only after further review. Comments:

1. If the tracheas were purchased at a butchery, why did the authors submit the study to an ethics committee?

2. There are no bibliographical references in the methodology section. It is true that some of the protocols used are referred in the discussion, but it would be convenient that the references were cited in each one of the sections of the methodology specifying if any adaptation of them has been made. This is especially important in points 2.3, 2.4, 2.5 and 2.6.

3. What breed of pig was used in this study?

4. In the methodology section, the distributors of all reagents used should be cited. If possible, it would be convenient to include their catalog numbers.

5. Regarding the evaluation of decellularization, measuring only the DNA concentration may not be enough. The size of the residual DNA is also relevant. Have the authors evaluated this by electrophoresis? On the other hand, the efficiency of decellularization is usually studied by DAPI and histological evaluation of the tissues subjected to the decellularization process. Why haven't the authors done this? This comment is relevant because the authors have used long segments of trachea and it could happen that decellularization was only effective at the ends of the segments.

6. Was the assessment of DNA content made from portions of the distal ends of the decellularized tracheas, or did they use central portion?

7. Related to the quantification of the percentage of collagen fibers, the authors should show representative panoramic and detail images of the analyzed histological sections. A greater detail in reference to how this quantification has been carried out is necessary.

8. In the discussion, in line 435 the authors use reference 30 to support evidence in mice kidney. However, this study is carried out with pig tracheas.

9. This is just a personal opinion, but considering the damage that Paolo Machiarni's case has caused to airway research, perhaps his studies (reference 3) should not be cited in honest papers like this one (https:/ /en.wikipedia.org/wiki/Paolo_Macchiarini)

Reviewer 2 Report

Genneral comments

The authors aimed to the development of a decellularization protocol of swine tracheas and tested different preservation methods of the tracheal scaffolds stored up to six months.

Findings showed that the storage protocols (PBS at 40C, alcohol at 40C, and slow cooling cryopreservation with cryopro-tectants), showed no significant differences in the amount of collagen and in the biomechanical properties of the scaffolds. The storage in PBS solution at 40C for six months did not change the scaffold mechanics.

The manuscript is well writen. I have only a few suggestions for the author to improve the article.

Specific comments

Material & Methods

For better visualization of the process, you could insert a study design figure

In line 229 and 358, Write temperatures with a space between the number and the degree. 4°C

Collagen quantification

In line 176, the histological process, were the slices fixed at 20% formaldehyde? The fixation processes are usually fixed in 4% paraformaldehyde.

No was performed analysis with Hematoxylin and Eosin (H&E), Masson’s Trichrome for observation the he organization and preservation of collagen fibers?

Results

Collagen level analysis. The confirmation of collagen presence must be performed using either immunohistochemistry or immunofluoresence assay against collagen type I. Also, the quantification of collagen with picrosirius is not accurate enough. The quantification of collagen should be performed using the hydroxyproline assay kit (e.g. using the kit MAK008, from Sigma Aldrich).

In line 250. in graph 4 could insert an image of picrosirus together to demonstrate the quantification of the collagen in natura and the others.

In line 216. Determination of tracheal and scaffold mechanical properties

In line 219. 8.18 ± 2.87 data is not in table, the review table and include p in text

Discussion

In line 270. Write in full acronym SD.

Round 2

Reviewer 1 Report

I want to thank the authors for their effort to answer my comments and congratulate them for the work done.